# *Chlamydomonas* ARMC2/PF27 is an obligate cargo adapter for intraflagellar transport of radial spokes

Karl F Lechtreck[1]*[†], Yi Liu[2†], Jin Dai[1], Rama A Alkhofash[1], Jack Butler[1], Lea Alford[3], Pinfen Yang[2]*

[1]Department of Cellular Biology, University of Georgia, Athens, United States; [2]Department of Biological Sciences, Marquette University, Milwaukee, United States; [3]Division of Natural Sciences,, Oglethorpe University, Atlanta, United States

**Abstract** Intraflagellar transport (IFT) carries proteins into flagella but how IFT trains interact with the large number of diverse proteins required to assemble flagella remains largely unknown. Here, we show that IFT of radial spokes in *Chlamydomonas* requires ARMC2/PF27, a conserved armadillo repeat protein associated with male infertility and reduced lung function. *Chlamydomonas* ARMC2 was highly enriched in growing flagella and tagged ARMC2 and the spoke protein RSP3 co-migrated on anterograde trains. In contrast, a cargo and an adapter of inner and outer dynein arms moved independently of ARMC2, indicating that unrelated cargoes distribute stochastically onto the IFT trains. After concomitant unloading at the flagellar tip, RSP3 attached to the axoneme whereas ARMC2 diffused back to the cell body. In *armc2/pf27* mutants, IFT of radial spokes was abolished and the presence of radial spokes was limited to the proximal region of flagella. We conclude that ARMC2 is a cargo adapter required for IFT of radial spokes to ensure their assembly along flagella. ARMC2 belongs to a growing class of cargo-specific adapters that enable flagellar transport of preassembled axonemal substructures by IFT.

**\*For correspondence:**
lechtrek@uga.edu (KFL);
pinfen.yang@marquette.edu (PY)

[†]These authors contributed equally to this work

**Competing interest:** The authors declare that no competing interests exist.

## Editor's evaluation

This paper is of broad interest to readers interested in motile cilia and cargo transport mediated by intraflagellar transport (IFT). It examines how radial spokes are trafficked into cilia by IFT, which represents a key process in the assembly of motile cilia. The authors demonstrate that an adaptor protein (ARMC2) is needed for association of radial spokes with the IFT machinery. They also find that three distinct axonemal proteins and adapters interact in a stochastic manner with individual IFT trains (or particles) rather than being transported by a specialized subset of trains specifically designated for axonemal proteins. The results support the key claims in the paper.

## Introduction

Cilia and eukaryotic flagella consist of hundreds of distinct proteins, which are synthesized in the cell body and moved posttranslationally into the organelle (*Rosenbaum and Child, 1967*; *Pazour et al., 2005*). Protein transport into flagella involves intraflagellar transport (IFT), a motor-based bidirectional motility of protein carriers (i.e., 'IFT trains') (*Kozminski et al., 1993*). Numerous proteins of the flagellar axoneme, matrix, and membrane have been shown to use the IFT pathway for flagellar entry and exit (*Lechtreck, 2015*). This raises the question how IFT trains, composed of just 22 IFT proteins and the associated kinesin-2 and IFT dynein motors, interact with the large number of diverse flagellar proteins. Tubulin, the most abundant flagellar protein, binds directly to the N-terminal domains of the

IFT-B core proteins IFT74 and IFT81 (*Bhogaraju et al., 2013*; *Craft et al., 2015*; *Kubo et al., 2016*; *Craft Van De Weghe et al., 2020*). However, many other proteins do not bind directly to the IFT trains but the interaction is mediated by IFT cargo adapters. The octameric BBSome, for example, acts as a linker for a diverse group of transmembrane and membrane-associated proteins, attaching them indirectly to IFT trains (*Nachury et al., 2007*; *Liu and Lechtreck, 2018*; *Wingfield et al., 2018*). With respect to axonemal proteins, IFT of outer dynein arms (ODAs) and the inner dynein arm (IDA) I1/f requires the adapter proteins ODA16 and IDA3, respectively (*Ahmed et al., 2008*; *Dai et al., 2018*; *Hunter et al., 2018*). The corresponding *oda16* and *ida3* mutants assemble full-length flagella that specifically lack ODAs or IDAs I1/f, respectively, but of otherwise normal ultrastructure (*Ahmed and Mitchell, 2005*; *Hunter et al., 2018*). ODAs and IDAs are large multiprotein complexes, which are assembled in the cell body before the entire substructures are moved into the flagella by IFT (*Fowkes and Mitchell, 1998*; *King, 2012*; *Viswanadha et al., 2014*). Similarly, more than 20 radial spoke (RS) proteins preassemble into a 12S RS precursor in the cell body (*Qin et al., 2004*; *Yang et al., 2006*). Then, the L-shaped precursors are moved by IFT to the flagellar tip, converted into the mature 20S spoke complexes, and assembled as T-shaped spokes onto the axonemal doublets (*Qin et al., 2004*; *Diener et al., 2011*; *Lechtreck et al., 2018*; *Grossman-Haham et al., 2021*; *Gui et al., 2021*). Mutations in the genes encoding the various spoke subunits lead to partial or complete loss of the RSs and flagellar paralysis (*Luck et al., 1977*; *Piperno et al., 1977*; *Witman et al., 1978*; *Piperno et al., 1981*; *Curry and Rosenbaum, 1993*). The *pf27* mutant, however, stands out as it assembles RSs of normal ultrastructure and subunit composition but the presence of spokes is limited to the very proximal region of the mutant flagella (*Huang et al., 1981*; *Alford et al., 2013*). In vitro decoration experiments using isolated axonemes and RSs revealed that the *pf27* axonemes bind control and *pf27* spokes, indicating that axonemal docking of RSs is unaffected in *pf27* (*Alford et al., 2013*). To explain the absence of spokes from large sections of the *pf27* flagella, *Alford et al., 2013*, postulated that *PF27* could encode a factor required for the transport of RSs into the distal flagellum via IFT. Then, RS assembly in the proximal region of *pf27* flagella could result from residual entry of RSs by diffusion followed by binding to the nearest available docking sites. Such a scenario could also explain why the phosphorylation state of several RS proteins is altered in *pf27* as it has been proposed that phosphorylation of these proteins occurs near the flagellar tip, which the RSs would fail to reach in *pf27* (*Huang et al., 1981*; *Yang and Yang, 2006*; *Gupta et al., 2012*). The *pf27* mutation maps close to the centromere of chromosome 12 but, despite whole genomes sequencing approaches, the *PF27* gene product remained unknown (*Kathir et al., 2003*; *Alford et al., 2013*).

Taking a candidate approach, we searched the region near the *pf27* locus for genes with a possible flagella-related function and identified *ARMC2*, encoding an armadillo repeat protein conserved in organisms with motile cilia (*Merchant et al., 2007*). The mammalian homologue of ARMC2 has been linked to reduced lung function and male infertility but the precise role of ARMC2 in the assembly of motile cilia remained unknown (*Soler Artigas et al., 2011*; *Coutton et al., 2019*; *Pereira et al., 2019*). A novel *Chlamydomonas armc2* mutant shares the RS-deficient phenotype of *p27*. Expression of ARMC2 restored wild-type motility and the presence of RSs in both *armc2* and *pf27* flagella, revealing that *PF27* encodes ARMC2. Fluorescent protein (FP)-tagged ARMC2 and the RS subunit RSP3 co-migrate on anterograde IFT trains in regenerating *Chlamydomonas* flagella whereas IFT of RSP3 was abolished in *armc2*. We conclude that ARMC2 is an adapter linking RSs to IFT to ensure their transport in flagella. Thus, IFT of ODAs, IDAs I1/f, and RSs, three large axonemal substructures pre-assembled in the cell body, requires adapters with single cargo specificity. As these adapters enter flagella and bind to IFT in the absence of their respective cargoes, we propose that the regulation of IFT-adapter interaction is a critical step for controlling cargo import into flagella by IFT.

## Results

### PF27 *encodes ARMC2*

The *pf27* locus was mapped to the midpoint between the *ODA9* and *TUB2* locus near the centromeric region of chromosome 12, placing it in the vicinity of the Cre12.g559250 gene, which encodes a 14-3-3 protein (*Kathir et al., 2003*). In the Phytozome genome browser (https://phytozome.jgi.doe.gov/pz/portal.html), we inspected this region for genes with a predicted role in flagella and identified Cre12.g559300 as a potential candidate. Cre12.g559300 encodes the armadillo repeat protein

ARMC2 (annotated as ARM1 in Phytozome), which is conserved in organisms with motile cilia (*Li et al., 2004*; *Merchant et al., 2007*). From the CLiP library we obtained the mutant strains LMJ. RY0402.155726 and LMJ.RY0402.083979, which have insertions in the 11th and last intron of Cre12. g559300, respectively (*Figure 1A*; *Li et al., 2019*). Strain LMJ.RY0402.155726 had paralyzed flagella displaying residual jerky movements resembling *pf27* and we refer to this strain as *armc2* (*Figure 1B*, *Figure 1—video 1*). The other strain (i.e., LMJ.RY0402.083979) was not analyzed further as it swam normally, which could potentially result from expression of a slightly truncated but apparently functional ARMC2 protein. Western blot analysis showed reduced levels of the RS proteins RSP3 and nucleoside diphosphate kinase 5 (NDK5 aka RSP23) in flagella of *armc2* and *pf27* (*Figure 1C, D* and *Figure 1—figure supplement 1A*). In comparison to wild-type flagella, the slower migrating phosphorylated forms of these proteins, which can be separated by long runs on 6% acrylamide gels, were less abundant in *armc2* flagella, as previously described for *pf27* (*Figure 1—figure supplement 1A*; *Huang et al., 1981*). To visualize the distribution of RSs in *armc2* flagella, we generated an *armc2 pf14* RSP3-NG strain by genetic crosses. *PF14* encodes the RS protein RSP3, which is critical for RS assembly and transport into flagella (*Diener et al., 1993*; *Lechtreck et al., 2018*). In control cells, RSP3-NG is present essentially along the length of flagella (*Figure 1—figure supplement 1B*; *Lechtreck et al., 2018*). In contrast, RSP3-NG was concentrated in the proximal region of the *armc2* flagella; it then tapered off and was largely absent from a large distal segment of flagella, similar to observations in *pf27* (*Figure 1E* a–f, F d–f, *Figure 1—figure supplement 1B*; *Alford et al., 2013*). In conclusion, *armc2* and *pf27* flagella share the same RS-related biochemical, structural, and functional defects.

Rarely, motile cells emerged in *armc2* and *pf27* cultures and, over a few days, the number of such cells increased, motility improved, and the amount of RSP3 in flagella increased (not shown). The phenomenon was not further explored in this study.

Using PCR, we engineered an *ARMC2* genomic construct encompassing the 11.3 kb *ARMC2* gene, ~1 kb of each of the 5' and the 3' flanking regions, and the aph7" selectable marker gene conferring resistance to hygromycin (*Figure 1—figure supplement 1C*). An NG-3xHA-6xHis tag, here referred to simply as 3xTAG, or an mScarlet (mS) tag was added upstream of the *ARMC2* Stop codon (*Figure 1— figure supplement 1C*). Transformation of *armc2* with the ARMC2-mS plasmid restored wild-type motility and Western blotting showed near wild-type levels of RSP3 in the flagella (*Figure 1B and C*). TIRF microscopy showed that the normal distribution of RSP3-NG along the length of flagella was reestablished in the *armc2 pf14* ARMC2-mS RSP3-NG rescue strain (*Figure 1E* g–i). Importantly, expression of ARMC2-3xTAG in *pf27* restored the presence and distribution of RSP3 in flagella and *pf27* ARMC2-3xTAG cells swam with wild-type motility (*Figure 1B and D*, F g–i). Thus, introduction of the *ARMC2* gene rescues the RS defects in both *pf27* and *armc2* indicating both mutants are allelic and that mutations in *ARMC2* underlie the *pf27* phenotype. Both mutant strains and the derived rescue strains were used to further characterize ARMC2 function.

## ARMC2/PF27 is a conserved 107-kD armadillo repeat protein

*Chlamydomonas ARMC2* is predicted to encode a 107 kD protein. Mass spectrometry of affinity-purified ARMC2-3xTAG was used to confirm the coding sequence of ARMC2. The purified fusion protein migrated at ~160 kD in Western blots, consistent with its predicted size (107 and 34 kD for the tag; *Figure 1—figure supplement 1D*). Using Ni-NTA-purification from whole cell extracts and anti-NG-nanobody trap purification from flagellar extracts, we identified a total of 17 unique ARMC2 peptides. Together with additional peptides obtained by mining other proteomic studies (*Zhao et al., 2019*; *Picariello et al., 2020*), the experimental peptides covered 42% of the predicted protein and were distributed through all but 1 (i.e., the 14th) of the 17 predicted ARMC2 exons (*Figure 1—figure supplement 1E*).

*Chlamydomonas* and human ARMC2 are reciprocal best hits in protein BLAST searches with an E value of 4E-50. The N-terminal ~400 residues of ARMC2 (~300 residues of the smaller human Armc2 isoform CRA_b) are predicted to be mostly intrinsically disordered by IUPred2A (IUPred2A (elte.hu)) and include the 10 phospho-sites identified by phosphoproteomics (*Figure 1—figure supplement 1E*, F) (*Wang et al., 2014*; *Erdős and Dosztányi, 2020*). The C-terminal ~600 residues of ARMC2 are predicted to be largely α-helical and encompass three armadillo repeats. Similar to previous efforts using whole genome sequencing, we failed to identify the causal genetic defect in *pf27* by sequencing

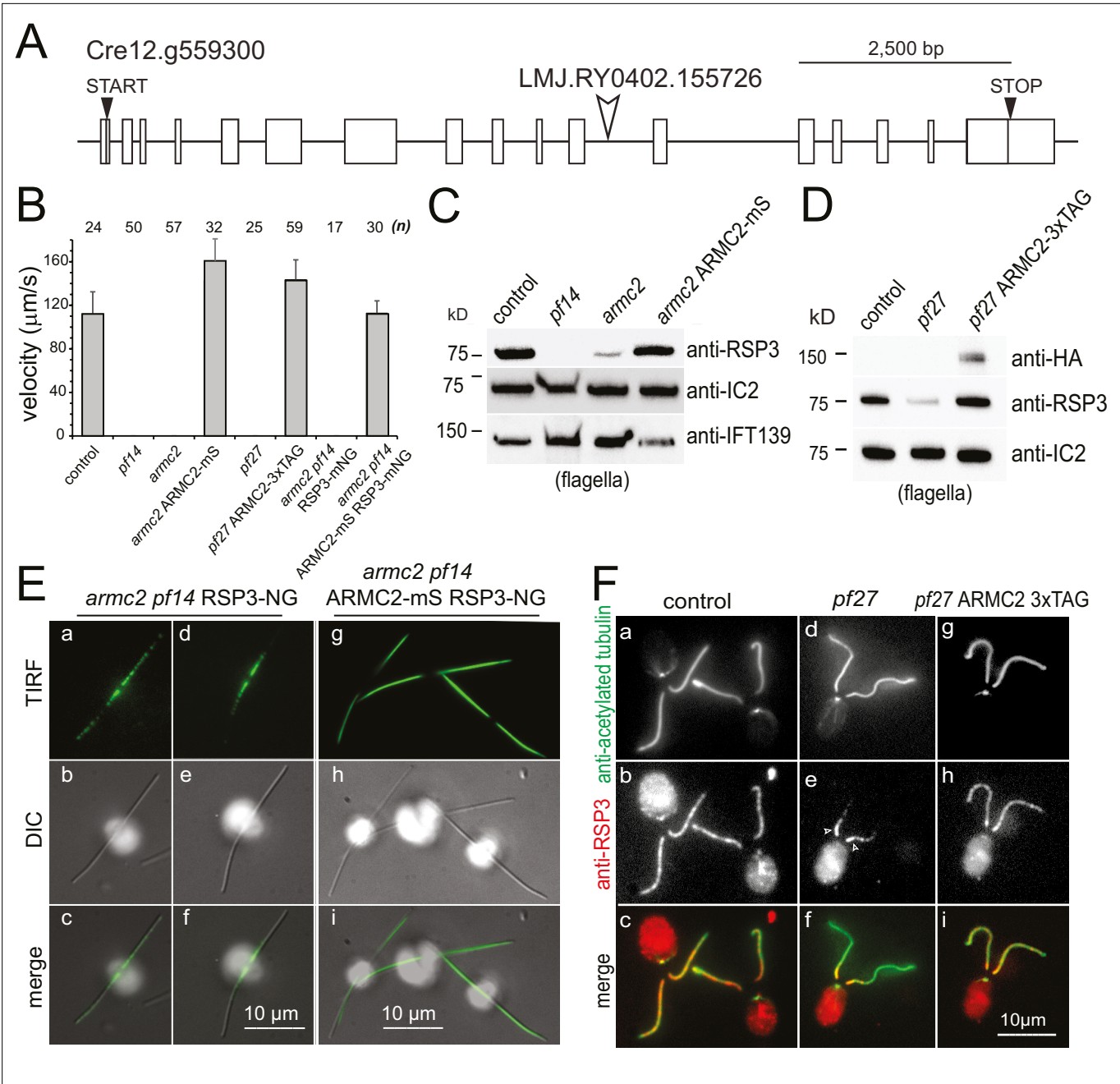

**Figure 1.** *PF27* encodes ARMC2. (**A**) Map of the *ARMC2* gene. The open arrowhead indicates the position of the insertion in the CLiP mutant LMJ. RY0402.155726. (**B**) Average swimming velocity of the strains as indicated. The standard deviation and the number of cells analyzed are indicated. (**C**) Western blot analysis of isolated flagella of control, the RSP3 mutant *pf14*, *armc2*, and the *armc2* ARMC2-mS rescue strain with antibodies to RSP3 and as loading controls, the outer arm dynein subunit IC2 and IFT139. Note accumulation of IFT139 in *pf14* and *armc2* as previously reported for paralyzed central pair mutants of *Chlamydomonas* (*Lechtreck et al., 2013*; see *Figure 2—figure supplement 1B*). (**D**) Western blot analysis of isolated flagella of control, *pf27*, and the *pf27* ARMC2-3xTAG rescue strain with antibodies to RSP3, anti-HA, and anti-IC2, as a loading control. Anti-HA was used to document expression of ARMC2-3xTAG (the 3xTAG encompasses a triple HA tag). A representative Western blot of three biological replicates is shown. (**E**) DIC and TIRF imaging of live cells showing the distribution of RSP3-NG in the *armc2 pf14* RSP3-NG mutant (a–f) and the *armc2 pf14* ARMC2-mS RSP3-NG rescue strain (g–i). Bars = 10 μm. (**F**) Immunofluorescence staining of methanol-fixed control (a–c), *pf27* (d–f), and *pf27* ARMC2-3xTAG (g–i) cells stained with anti-acetylated-α-tubulin (a, d, g) to visualize flagella and affinity-purified anti-RSP3 (b, e, h); merged images are shown in the bottom row (c, f, i). Arrowheads in e, residual RSP3 near the proximal end of the *pf27* flagella. The bright signal of the cell body likely results from unspecific binding of the anti-RSP3 antibody and chlorophyll autofluorescence. Bar = 10 μm.

The online version of this article includes the following video and figure supplement(s) for figure 1:

*Figure 1 continued on next page*

*Figure 1 continued*

**Figure supplement 1.** Cloning, mass spectroscopy, and structure of *Chlamydomonas* ARMC2.

**Figure 1—video 1.** *armc2* has a paralyzed flagella phenotype.

https://elifesciences.org/articles/74993/figures#fig1video1

of genomic PCR products; a possible contributing factor is the highly repetitive nature of the centromeric DNA.

## ARMC2 is highly enriched in regenerating flagella

In the *pf27* and *armc2* mutants, RS assembly onto the axoneme is rather incomplete but for the proximal region of the flagella (*Alford et al., 2013*). To address the question how ARMC2/PF27 promotes RS assembly along the length of flagella, we turned to TIRF imaging of ARMC2-3xTAG expressed in the *pf27* mutant. In full-length flagella, we observed only few ARMC2-3xTAG particles moving by diffusion and, occasionally, by anterograde IFT (0.9 IFT events/min/flagellum, SD 1.6 events/min/flagellum, n = 16; *Figure 2Aa*, B). In contrast, anterograde IFT of ARMC2-3xTAG was frequent in regenerating flagella with an average transport frequency of 44 events/min/flagellum (SD 16.2 events/min/flagellum, n = 34) approaching those observed for GFP-tagged IFT itself (~60–80/min; *Figure 2Ab*, B) (*Kozminski et al., 1993*; *Reck et al., 2016*; *Wingfield et al., 2017*). Typically, ARMC2-3xTAG moved in one processive run from the flagellar base to the tip by anterograde IFT with an average velocity of 1.71 µm/s (SD 0.24 µm/s, N = 65 particles; green arrowheads in *Figure 2C* a–c). As previously described for IFT proteins and other cargoes, ARMC2-3xTAG dwelled at the flagellar tip on average for 2.3 s (SD 1.8 s, n = 67; white brackets in *Figure 2C* a–c) (*Wren et al., 2013*; *Chien et al., 2017*). Dwelling and high-frequency transport resulted in the formation of a pool of ARMC2-3xTAG at the tip of regenerating flagella (*Figure 2—figure supplement 1A*). Once released from IFT, ARMC2-3xTAG diffused swiftly into the flagellar shaft (white arrowheads in *Figure 2C* a–c; *Figure 2—video 1*). Retrograde IFT of ARMC2-3xTAG was rare (*Figure 2B*, red arrow in *Figure 2C* d) indicating that the protein returns mostly by diffusion to the cell body as previously described for the anterograde IFT motor kinesin-2 and IDA3, the cargo adapter for IDA I1/f (*Chien et al., 2017*; *Hunter et al., 2018*). Typically, ARMC2-3xTAG bleached in a single step indicating that only a single copy of the tagged protein was present on an individual IFT train (dashed circle in *Figure 2C* b). However, trains carrying two copies of ARMC2-3xTAG were also observed (*Figure 2C* e). Similar to other proteins transported by IFT, ARMC2-3xTAG displayed a variety of less frequent behaviors such as unloading from anterograde trains along the length of the flagella and re-binding to subsequent trains (not shown).

To determine the degree of ARMC2-3xTAG accumulation in growing flagella, we isolated full-length and regenerating flagella from the *pf27* ARMC2-3xTAG strain for Western blot analysis (*Figure 2D and E*). When a similar number of flagella, that is, one regenerating for each full-length flagellum, were loaded, ARMC2-3xTAG was enriched about 14× in the regenerating flagella (*Figure 2E*, lanes 1 and 2; *Figure 2—figure supplement 2*). In contrast, the IFT particle protein IFT54 was similarly abundant in both samples, in agreement with previous observations that the amount of IFT proteins in flagella is largely independent of flagellar length (*Marshall et al., 2005*). When approximately similar amounts of protein were loaded, that is, several short flagella for each full-length flagellum, both IFT54 and especially tagged ARMC2 were enriched in regenerating flagella (*Figure 2E*, lanes 3 and 4). Thus, ARMC2 is highly enriched in growing flagella.

To test if IFT of ARMC2 depends on the presence of intact RSs, we imaged ARMC2-3xTAG in a *pf14 armc2* double mutant. ARMC2-3xTAG moved by IFT in *pf14* flagella and the frequency of its transport was upregulated in regenerating *pf14* flagella (*Figure 2A* c and d, B). In conclusion, IFT of ARMC2-3xTAG and its regulation by flagellar length do not require the presence of RSP3. IFT of ARMC2-3xTAG during flagellar regeneration in the *pf14* background, however, was notably less frequent than in cells with intact RSs (*Figure 2A and B*) and Western blot analysis of whole cell samples showed similar amounts of ARMC2-3xTAG in the *pf27* ARMC2-3xTAG rescue strain and the *pf14 armc2* ARMC2-3xTAG strain (*Figure 2F*). Thus, the presence of intact RSs could promote IFT of ARMC2-3xTAG. However, similar to other strains with large structural defects in the axoneme, *pf14* flagella regenerate slower than those of wild-type strains and accumulated IFT proteins (*Figure 1C*, *Figure 2—figure supplement 1B* and not shown) (*Lechtreck et al., 2013*). Therefore, reduced IFT of

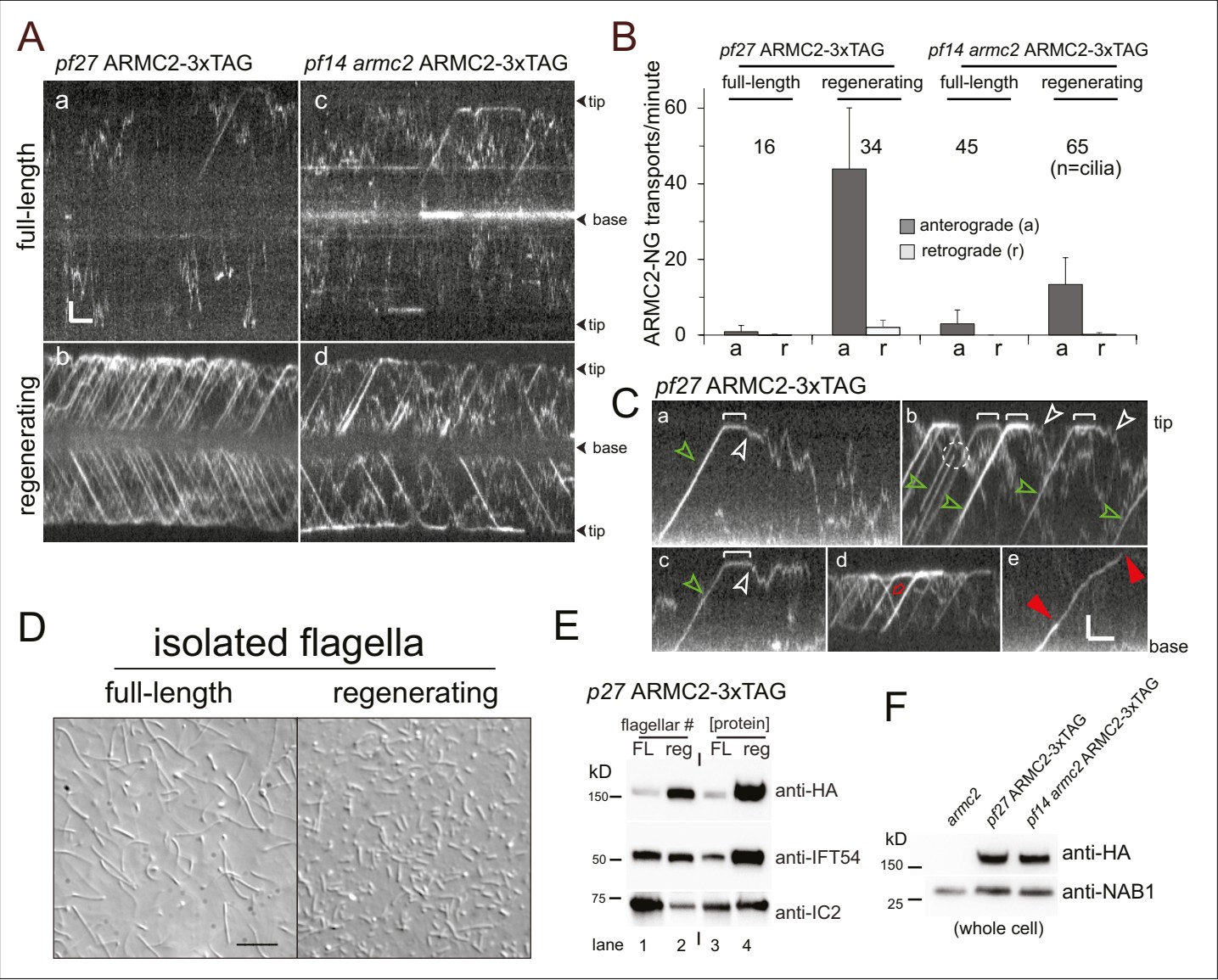

**Figure 2.** ARMC2-3xTAG is highly enriched in regenerating flagella. (**A**) TIRF imaging of ARMC2-3xTAG in the *pf27* background (**a, b**) and the *pf14 armc2* double mutant background (**c, d**) in full-length (**a, c**) and in regenerating flagella (**b, d**). Bars = 2 s 2 µm. The flagellar tips and bases are indicated. (**B**) Bar graph showing the average frequencies (events/min/flagellum) of anterograde and retrograde transport of ARMC2-3xTAG in full-length and regenerating flagella of the *pf27* ARMC2-3XTAG and the *pf14 armc2* ARMC2-3xTAG strain. The standard deviation and the number of flagella analyzed are indicated. (**C**) Kymograms of ARMC2-3xTAG in late regenerating *pf27* flagella. The white brackets in a–c mark the dwell time of individual ARMC2-3xTAG particles between arrival at the tip by anterograde IFT and the onset of diffusion (white arrowheads). Green arrowheads in a–c, anterograde transport of ARMC2-3xTAG, red open arrow in d, retrograde IFT of ARMC2-3xTAG; red arrowheads in e, stepwise bleaching of ARMC2-3xTAG indicating for the presence of two copies. In c, a single step bleaching event is marked by a dashed circle. Bars = 2 s and 2 µm. (**D**) DIC images of full-length and regenerating flagella of the *pf27* ARMC2-3xTAG strain. Regenerating flagella were harvested ~22 min after deflagellation by a pH shock. Bar = 10 µm. (**E**) Western blot analysis of the full-length and regenerating flagella shown in C with the antibodies indicated. On the left side, an equal number of flagella were loaded and on the right side, approximately equal loading of protein was attempted. (**F**) Western blot comparing the presence of ARMC2-3xTAG in the *pf27* ARMC2-3xTAG and the *pf14 armc2* ARMC2-3xTAG strain. Antibodies to the cell body protein nucleic acid binding protein 1 (NAB1) were used as a loading control.

The online version of this article includes the following video and figure supplement(s) for figure 2:

**Figure supplement 1.** ARMC2-3xTAG accumulates at the tip of growing flagella.

**Figure supplement 2.** ARMC2-3xAG is enriched in growing flagella.

**Figure 2—video 1.** Anterograde intraflagellar transport (IFT) of ARMC2-3xTAG.

https://elifesciences.org/articles/74993/figures#fig2video1

ARMC2-3xTAG in *pf14* could also result from a more general imbalance of IFT rather than from the absence of intact spokes.

## RSP3-NG co-migrates with ARMC2-mS during anterograde IFT

In growing flagella, ARMC2-3xTAG moves on anterograde IFT trains suggesting that it could assist in IFT of RSs, which are also transported more frequently during flagellar growth (*Figure 3A and B*; *Lechtreck et al., 2018*). Two-color TIRF imaging of ARMC2-mS and RSP3-NG in the *armc2 pf14* double-mutant-double-rescue strain revealed co-migration of the two tagged proteins during anterograde IFT (*Figure 3A*, *Figure 3—video 1*). In detail, 104 (80%) of 130 observed RSP3-NG anterograde transports observed over 1622 s in 20 regenerating flagella were accompanied by ARMC2-mS (*Figure 3A and C*, *Table 1*). In contrast, ~82% of the ARMC2 transport were not accompanied by an RSP3-NG signal showing that ARMC2-mS transports were considerably more frequent than those of RSP3-NG in both full-length and regenerating flagella (*Figure 3A and C*, *Table 1*). With respect to RSP3-NG transports lacking an ARMC2-mS signal, we note that in order to visualize IFT of RSP3-NG, we first had to photobleach the substantial amounts of RSP3-NG already incorporated into the axoneme. As mS is less photostable than NG, it is likely that some ARMC2-mS was photobleached during this preparatory step probably explaining the occurrence of some RSP3-NG transports without a co-migrating ARMC2-mS signal. To conclude, RSP3-NG co-migrates with ARMC2-mS during anterograde IFT but the transport frequency of ARMC2-mS exceeds that of its RSP3-NG.

To test whether RSP3-NG moves by IFT in cells lacking ARMC2, we expressed RSP3-NG in an *armc2 pf14* double mutant and analyzed full-length and regenerating flagella (*Figure 3D*). Some RSP3-NG particles moved inside flagella by diffusion and, as flagella elongated, the amount of RSP3-NG anchored in the proximal flagellar region increased (*Figure 3D* d–j) (*Alford et al., 2013*). But for rare ambiguous events, transport of RSP3-NG by IFT was not observed in the *armc2* mutant background (*Figure 3D*). We conclude that ARMC2 is required for IFT of RSs.

ARMC2 is largely absent from full-length flagella while RSs are retained, indicating that the two will separate after cotransport. To analyze the behavior of the ARMC2-mS RSP3-NG complexes at the tip, we focused on the later stages of flagellar regeneration (>30 min after pH shock) when ARMC2-mS traffic was less dense increasing the chance of observing individual ARMC2-mS particles. Further, we photobleached the tip of the regenerating flagella in a subset of experiments to prevent unbleached RSP3-NG from accumulating as it incorporates into the elongating axoneme. The analysis of 15 such ARMC2-mS RSP3-NG complexes showed that the release of ARMC2-mS from its dwell phase at the tip occurred concurrently with the onset of RSP3-NG movements (white arrowheads in *Figure 4*). While ARMC2-mS mostly diffused into the flagellar shaft, RSP3-NG moved to a somewhat more subdistal position, where it typically remained stationary for extended periods of time (12 of 15 events) potentially indicating stable docking to the axoneme. Diffusion of RSP3-NG deeper into the flagellar shaft (2 of 15 events) or return by retrograde IFT (1 of 15 events) was also observed (not shown). Thus, ARMC2 and RSP3 separate after arrival and dwelling at the tip with RSP3 remaining in the flagellum and ARMC2 returning to the cell body.

## The IFT frequency of ARMC2 is regulated by flagellar length

The frequency of ARMC2-3xTAG transport by IFT is upregulated in short growing flagella. To gain insights into how the frequency of ARMC2 transport is regulated, we generated long-zero cells by removing just one of the two flagella of a given cell by mechanical shearing. Such long-zero cells will regrow the missing flagellum while shortening the remaining one until both flagella are of approximately the same length (*Rosenbaum et al., 1969*; *Ludington et al., 2012*). Then, both flagella will regrow to full length. In all 19 *pf27* ARMC2-3xTAG long-short cells analyzed, the IFT frequency of ARMC2-3xTAG in the shorter flagellum exceeded that of the longer flagellum (on average 29.1 events/min, SD 16 events/min for the shorter vs. 8.9 events/min, SD 6.8 events/min for the longer flagellum; *Figure 5A and B*; *Figure 5—video 1*). As the length difference between the short and long flagellum decreased, the difference in ARMC2-3xTAG transport frequency between the two flagella also diminished (*Figure 5A* c and d, B). A similar behavior, that is, flagellum-autonomous regulation of transport frequency by flagellar length, was previously documented for GFP-tubulin, DRC4-GFP, and the cargo adapter IDA3 (*Craft et al., 2015*; *Hunter et al., 2018*).

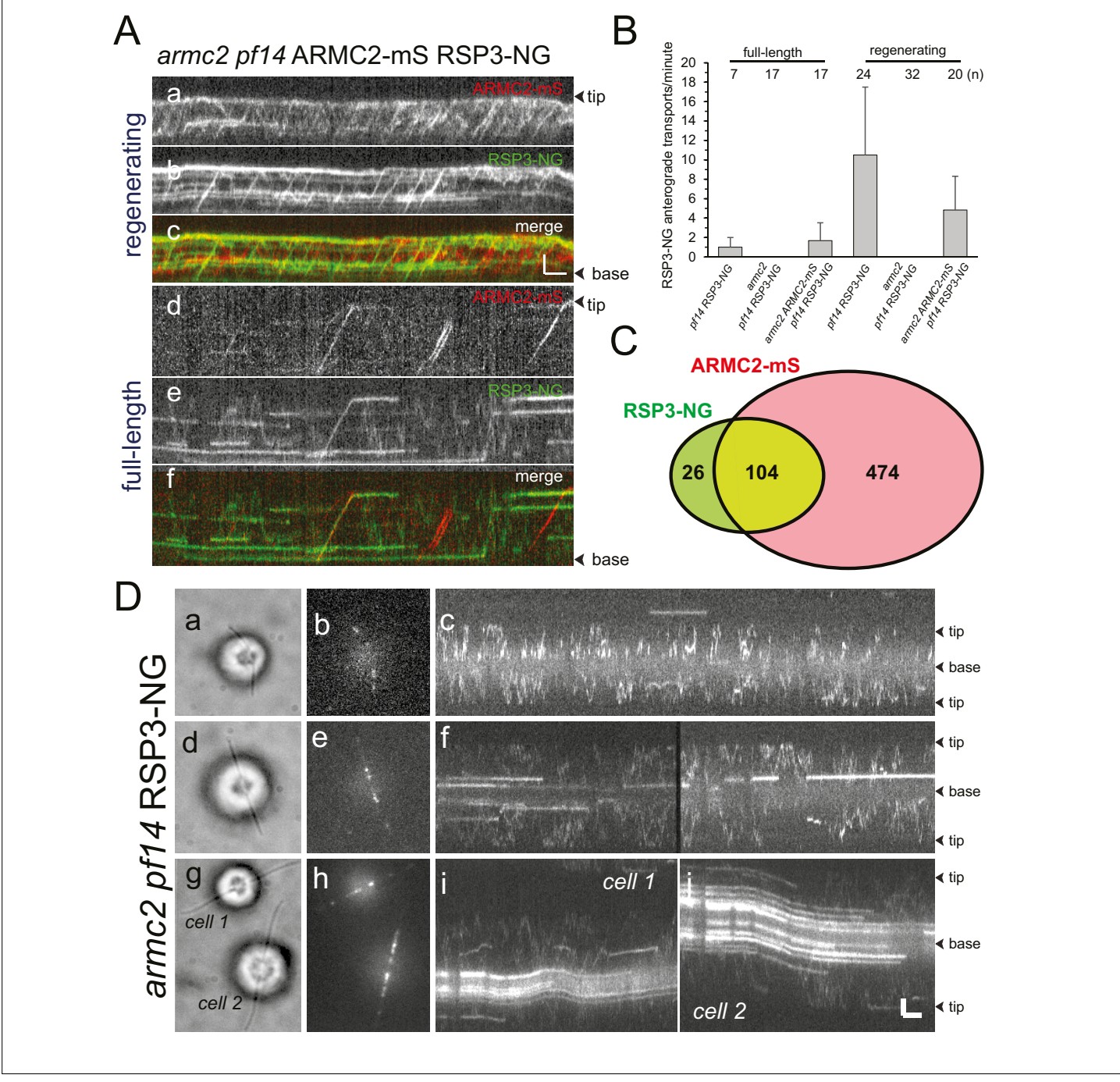

**Figure 3.** ARMC2-mS and RSP3-NG co-migrate by anterograde intraflagellar transport. (**A**) Two-color TIRF imaging of a regenerating (a–c) and a full-length (d–f) flagellum of the *armc2 pf14* ARMC2-mS RSP3-NG strain. Horizontal trajectories result from residuals unbleached RSP3-NG in the axoneme. Bars = 2 s 2 µm. (**B**) Bar graph showing the frequencies (events/min/flagellum) of RSP3-NG transports by anterograde IFT in full-length and regenerating flagella of the *pf14* RSP3-NG, *armc2 pf14* RSP3-NG, and *armc2 pf14* ARMC2-mS RSP3-NG strains. The standard deviation and the number of flagella analyzed are indicated. (**C**) Venn diagram showing the distribution of anterograde ARMC2-mS and RSP3-NG transports; the overlap area represents the cotransports corresponding to 82% of all RSP3-NG and 18% of the ARMC2-mS transports. (**D**) IFT of RSP3-NG requires ARMC2. Analysis of RSP3-NG in *armc2* mutant flagella. Brightfield (**a, d, g**) and TIRF (**b, e, h**) still images and corresponding kymograms (**c, f, i, and j**) of early (**a–c**), mid (**g–f**), and late stage (**g–j**) regenerating *armc2 pf14* RSP3-NG cells. Bars = 2 s and 2 µm.

The online version of this article includes the following video for figure 3:

**Figure 3—video 1.** Cotransport of RSP3-NG and ARMC2-mS.

https://elifesciences.org/articles/74993/figures#fig3video1

**Table 1.** Frequency of cargo cotransport.

The table list the observed anterograde transports for ARMC2-mS, RSP3-NG, IDA3-NG, and IC2-NG in flagella of the corresponding double-mutant-double-rescue strains. For calculating the probability, by which an intraflagellar transport (IFT) train carries a cargo, we assumed an IFT frequency of 1/s. For cotransports, we calculated the observed probability (cotransports/total time) and compared it to the probability of cotransports occurring by chance as calculated by the following formula: P(cotransport–calculated) = P(cargo A) × P(cargo B).

| Strain | ARMC2-mS (n) | RSP3-NG (n) | Cotransports (n) | Time (s) | P(ARMC2-mS) | P(RSP3-NG) | P(cotransports-observed) | P(cotransport-calculated) | Cilia (n) |
|---|---|---|---|---|---|---|---|---|---|
| pf14 armc2RSP3-NGARMC2-mS(full length) | 125 | 42 | 26 | 1504 | 0.083 | 0.028 | 0.017 | 0.0023 | 17 |
| pf14 armc2RSP3-NGARMC2-mS(regenerating) | 578 | 130 | 104 | 1622 | 0.36 | 0.08 | 0.064 | 0.029 | 20 |

| | ARMC2-mS (n) | IDA3-NG (n) | Cotransports (n) | Time (s) | P(ARMC2-mS) | P(IDA3-NG) | P(cotransports-observed) | P(cotransport-calculated) | Cilia (n) |
|---|---|---|---|---|---|---|---|---|---|
| ida3 armc2IDA3-NGARMC2-mS(regenerating) | 243 | 106 | 20 | 905 | 0.27 | 0.12 | 0.022 | 0.03 | 35 |

| | ARMC2-mS (n) | IC2-NG (n) | Cotransports (n) | Total time (s) | P(ARMC2-mS) | P(IC2-NG) | P(cotransports-observed) | P(cotransport-calculated) | Cilia (n) |
|---|---|---|---|---|---|---|---|---|---|
| oda3 oda armc2 ODA6-NG ARMC2-mS (regenerating) | 82 | 78 | 3 | 1575 | 0.052 | 0.049 | 0.0019 | 0.0026 | 20 |

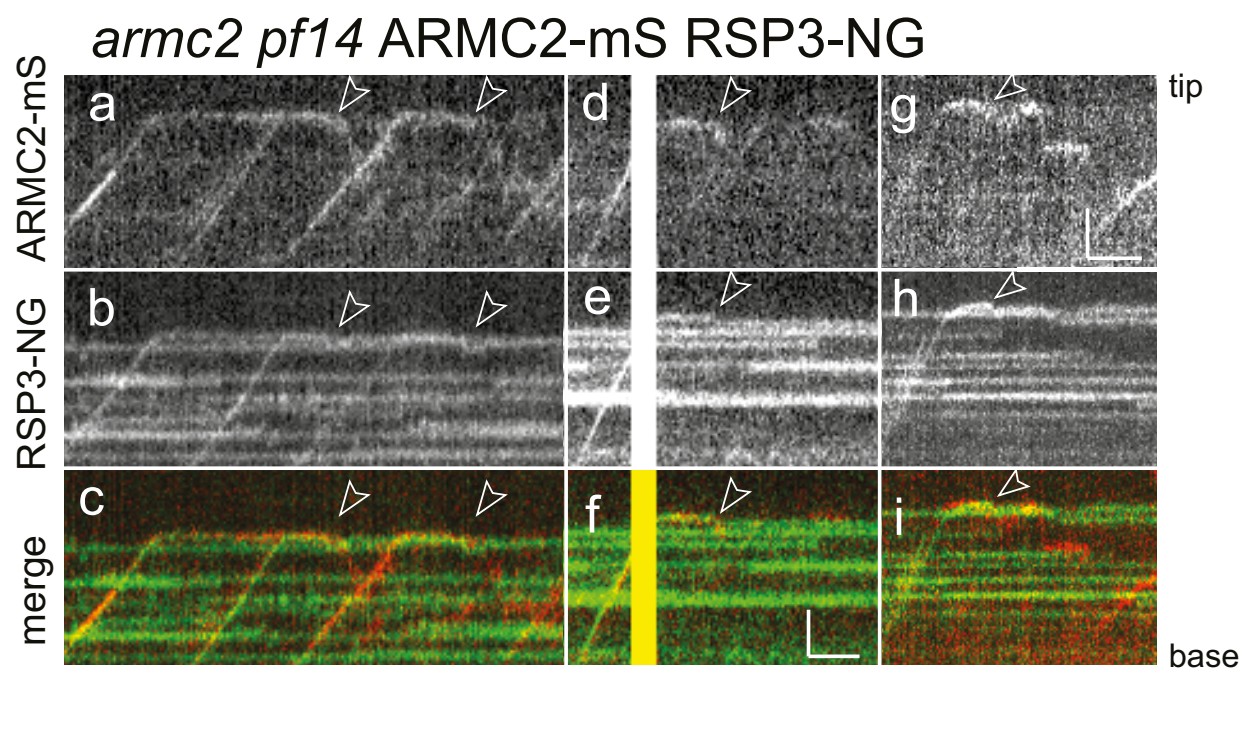

**Figure 4.** RSP3-NG ARMC2-mS complexes dissociate at the flagellar tip. Kymograms from simultaneous imaging of the cargo adapter ARMC2-mS (**a, d, g**) and its cargo RSP3-NG (**b, e, h**); the merged images are shown in c, f, and i. The end of the dwell phase and concomitant onset of ARMC2-mS and RSP3-NG movements are marked with white arrowheads. The white/yellow frames in d–f result from overexposure due to the use of the bleaching laser pointed at the other flagellar tip of the cell. Bars = 2 s and 2 µm.

To test if the upregulation of ARMC2 transport during flagellar regeneration depends on de novo protein synthesis, we incubated cells in the protein synthesis inhibitor cycloheximide (CHX; 10 µg/ml) for 1 hr prior to generating long-short cells followed by regeneration in CHX-containing medium (*Figure 5C*). The frequency of anterograde ARMC2-3xTAG transport in the shorter flagellum exceeded that of the longer flagellum of the same cell in all 14 long-short cells analyzed with average transport frequencies of 33 events/min (SD 13.2 events/min, n = 14) and 10.2 events/min (SD 8.1 events/min, n = 14) for short and long flagella, respectively (*Figure 5C*, *Figure 5—video 2*). Thus, ARMC2-3xTAG anterograde transport in regenerating short flagella is approximately 4× more frequent than in the longer flagella of the same cell regardless of the cycloheximide treatment.

We also deflagellated cells by a pH shock in the presence of CHX and allowed them to regrow flagella in the presence of 10 µg/ml CHX for 90 min, at which point the flagella are approximately half-length and elongation had ceased due to the absence of protein synthesis (*Rosenbaum et al., 1969*). In such cells, the frequency of ARMC2-3xTAG transport was 14.4 events/min (SD 4.3, n = 10) well above the frequency determined for full-length flagella (see *Figure 2B*). Thus, high-frequency ARMC2-3xTAG transport is indeed triggered by the insufficient length of the flagella rather than active growth of flagella. Taken together, the data indicate that cells possess a pool of ARMC2-3xTAG and that ARMC2-3xTAG will preferably attach to IFT trains that enter short flagella.

### The ARMC2 cell body pool

The distribution and dynamics of cargo adapters and cargo proteins in the cell body remains largely unknown. The data above established the presence of an ARMC2-FP pool in the cell body. To determine its distribution, we analyzed *pf14 armc2* RSP3-NG ARMC2-mS cells using epifluorescence, which revealed the presence of an ARMC2-mS pool near the flagellar base of most cells (*Figure 5—figure supplement 1A*). To analyze the dynamics of the ARMC2 pool, we increased the incidence angle of the TIRF laser, allowing us to image ARMC2-3xTAG positioned deeper in the cell body (*Figure 5—figure supplement 1B*). The ARMC2-3xTAG pool typically consisted of two closely spaced dots and

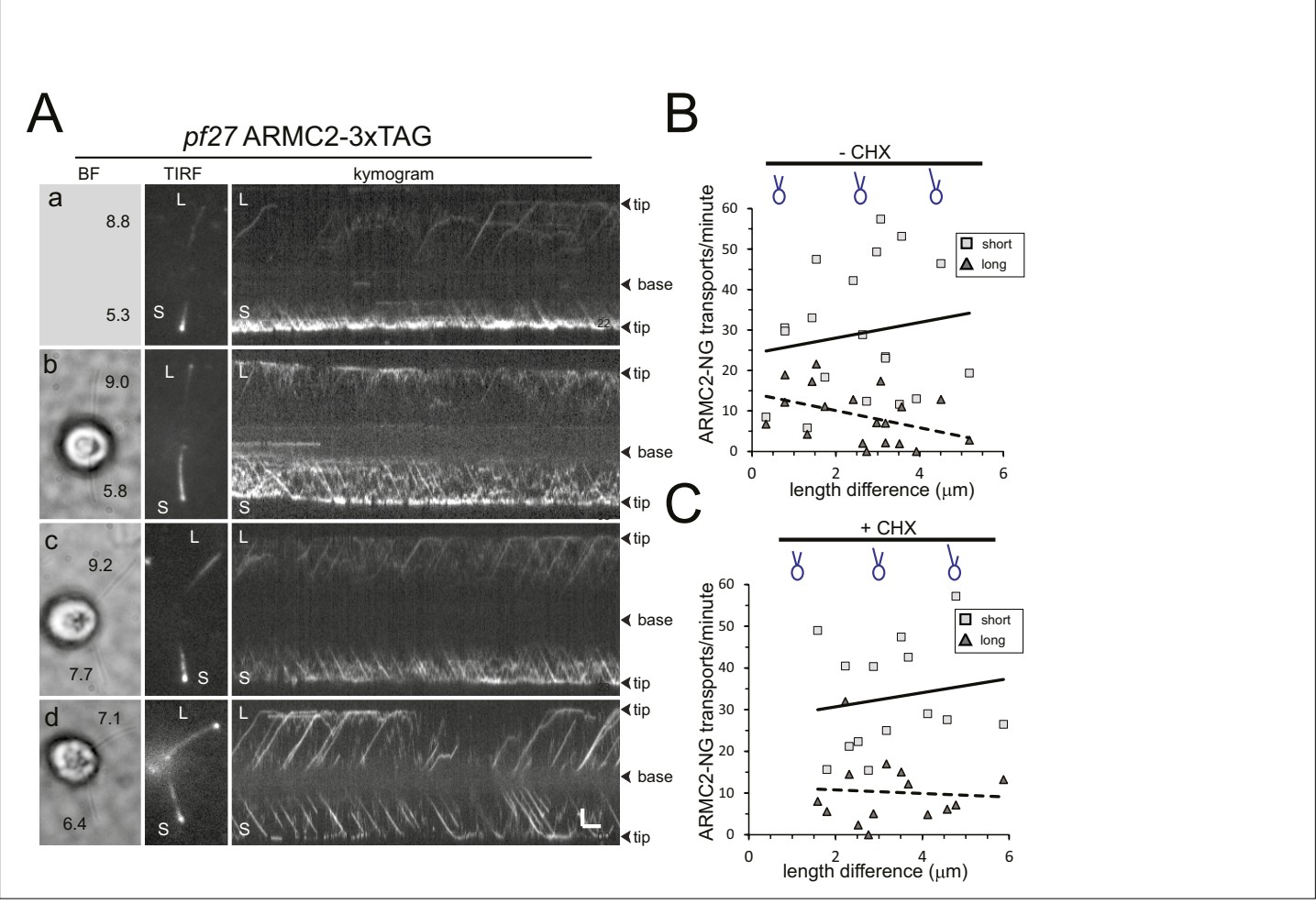

**Figure 5.** ARMC2-3xTAG transport is upregulated in short flagella. (**A**) Gallery of brightfield (BF) and TIRF still images and the corresponding kymograms of long-short p27 ARMC2−3xTAG cells. No BF image was recorded for the cell shown in a. The length of the long (**L**) and short (**S**) flagella is indicated (in µm in a–d). Bars = 2 s and 2 µm. (**B**) Plot of the ARMC2-3xTAG transport frequency (events/min/flagellum) in the short (squares) and the long (triangles) flagella against the length difference between the two flagella. Trendlines, solid for the long and dashed for the short flagella, were added in Excel. (**C**) As B, but for cells treated with cycloheximide prior and during the experiment.

The online version of this article includes the following video and figure supplement(s) for figure 5:

**Figure supplement 1.** ARMC2-FP forms a pool near the basal bodies.

**Figure 5—video 1.** ARMC2-3xTAG transport in long-short cells.

https://elifesciences.org/articles/74993/figures#fig5video1

**Figure 5—video 2.** ARMC2-3xTAG transport in long-short cells.

https://elifesciences.org/articles/74993/figures#fig5video2

we use a focused laser beam to photobleach one of the two dots (*Figure 5—figure supplement 1B* and C). Due to the closeness of the two dots, the signal of the second dot was diminished during the bleaching step but could still be used as a control to estimate signal recovery. Analysis of a small number of *pf27* ARMC2-3xTAG and *pf14 armc2* ARMC2-3xTAG cells (n = 3) showed that recovery of the ARMC2-3xTAG was incomplete and plateaued after ~20 s suggesting a slow replacement of the bleached protein in the pool.

ARMC2-3xTAG cells with regenerating flagella were used to determine the dwell time of ARMC2 in the basal body-associated pool. After bleaching of the entire basal body-associated pool, IFT of ARMC2-3xTAG was interrupted for ~19 s (SD 7.7 s, n = 12, red bracket in *Figure 5—figure supplement 1D*, top panel). A similar average length of the gap was observed in the *pf14 armc2* ARMC2-3xTAG (20 s, SD 10.7 s, n = 18; *Figure 5—figure supplement 1D*, bottom panel) suggesting that

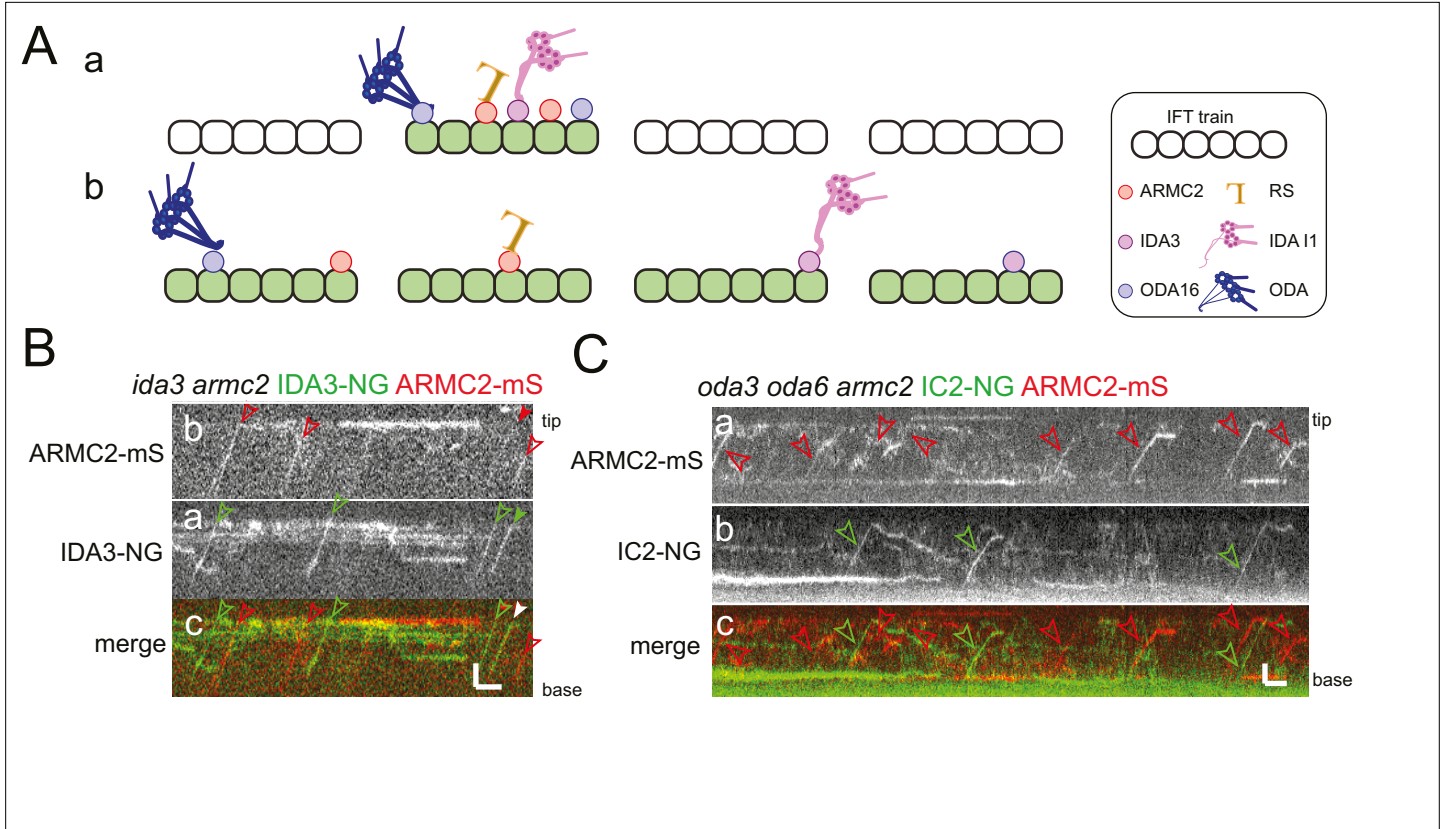

**Figure 6.** ARMC2-mS is transported independently of IDA3-NG and IC2-NG. (**A**) Schematic presentation of two models for intraflagellar transport (IFT)-cargo interaction using radial spokes (RSs), outer dynein arms (ODAs), and I1 inner dynein arms (IDAs) and their adapters as examples. (**a**) Most cargoes use a specific subset of IFT trains, which have a high propensity to bind axonemal proteins, for example, because they are in a hypothetical open configuration. (**b**) All IFT trains are similarly capable of binding axonemal cargoes; thus, cargoes are stochastically distributed onto the trains. (**B**) Kymograms of two-color TIRF imaging of ARMC2-mS and IDA3-NG in an *ida3 armc2* mutant cell. ARMC2-mS trajectories are marked with open red arrowheads and IDA3-NG trajectories with open green arrowheads. Filled arrowheads indicate a cotransport. Bars = 2 s and 2 μm. (**C**) Kymograms of two-color TIRF imaging using the *oda3 oda6 armc2* IC2-NG ARMC2-mS strain. Trajectories of ARMC2-mS and IC2-NG transports are marked with red and green arrowheads, respectively. Bars = 2 s and 2 μm.

ARMC2 dynamics in the pool, that is, the rate by which it travels through the pool and attaches to IFT, are not altered by the absence of its cargo. After the gap, ARMC2-3xTAG traffic recommenced albeit at a reduced frequency in agreement with the incomplete recovery of the basal body signal (*Figure 5—figure supplement 1D*). Considerably shorter gaps were observed for several IFT proteins (~6 s or less) and tubulin-GFP in similar bleaching experiments (~2 s) (*Wingfield et al., 2017*). Thus, ARMC2-3xTAG dwells in the basal body-associated pool considerably longer than the IFT proteins of the train, which will carry it into the cilium, suggesting that IFT proteins and ARMC2 are recruited through different routes.

## ARMC2, IDA3, and IC2 are stochastically distributed onto IFT trains

Similar to previous observations on the I1/f transport adapter IDA3 and several axonemal proteins (*Wren et al., 2013*; *Craft et al., 2015*; *Hunter et al., 2018*), IFT of ARMC2-3xTAG is frequently observed in short flagella but progressively decreases as flagella approach full length. How axonemal proteins and their adapters are distributed onto the IFT trains remains unknown as the currently available cryo-EM structures of IFT trains were obtained by image averaging, which will cancel out signals from substoichiometric train components such as cargoes (*Jordan et al., 2018*). One possible model is that a subset of trains is specialized to carry axonemal cargoes, for example, because they are in an 'open' state allowing cargoes to adhere. Such 'axonemal cargo trains' could frequently enter short flagella but only sporadically move into full-length flagella explaining the flagellar length-dependent decline in cargo IFT. The model predicts that different axonemal proteins preferentially travel on

this subclass of trains. Alternatively, all trains could be equally capable of binding cargo and distinct cargoes will distribute stochastically onto the trains (*Figure 6A*).

We generated strains expressing the adapter proteins ARMC2-mS and IDA3-NG in a corresponding *armc2 ida3* double mutant and expressing ARMC2-mS together with the essential ODA subunit IC2-NG in an *oda3 armc2 oda6* strain (*Figure 6B and C*). IC2-NG rescues the ODA-deficient *oda6* mutant and the ODA docking complex-deficient *oda3* mutant background was chosen because it interferes with the binding of ODAs to the axoneme preventing the accumulation of IC2-NG in flagella and thereby circumventing the need to photobleach the flagella prior to imaging (*Koutoulis et al., 1997*; *Dai et al., 2018*). Live imaging of ODA16, the adapter required for ODA transport, has not yet been achieved. We focused on mid to late regenerating cells when the transport of these proteins was infrequent, reducing the probability that both proteins are present on a given train by chance. In 35 regenerating *armc2 ida3* ARMC2-mS IDA3-NG flagella analyzed over 905 s, we observed 243 ARMC2-mS and 106 IDA3-NG particles moving anterogradely by IFT, of which 20 were cotransported on the same trains (*Table 1*). For the *oda3 armc2 oda6* ARMC2-mS ODA6-NG strain, we analyzed 20 flagella for 1575 s and observed 82 ARMC2-mS and 78 IC2-NG particles, of which three were cotransports (*Table 1*). The lower frequency of ARMC2-mS transports in the latter experiment could reflect that more cells in later stages of regeneration were analyzed as indicated by the higher average length of cilia (8.3 μm ± 1.85 μm vs. 7.3 μm ± 1.5 μm). As anterograde IFT has a frequency of ~60 trains/min (*Kozminski et al., 1993*; *Dentler, 2005*; *Engel et al., 2012*; *Reck et al., 2016*), the observed probability that a train carries both cargoes was 0.022 for ARMC2-mS and IDA3-NG (i.e., ~2% of the trains carry ARMC2-mS and IDA3-NG) and 0.0019 for ARMC2-mS and IC2-NG. These values are close to the calculated probabilities of 0.03 for the ARMC2-mS/IDA3-NG pair and 0.0026 for the ARMC2-mS/IC2-NG pair by which such cotransports would occur by chance if the two cargoes are transported independently of each other without a preference for a subclass of IFT trains (see *Table 1* and Materials and methods for calculation). In contrast, RSP3-NG transport depends on ARMC2-mS and as expected ARMC2-mS and RSP3-NG were cotransported with a probability of 0.027 in full-length flagella significantly exceeding the value of 0.0023 calculated if these two proteins would attach independently of each other to the trains (*Table 1*). In conclusion, ARMC2-mS, IDA3-NG, and IC2-NG are transported independently of each other arguing against a subclass of IFT trains specialized to carry axonemal proteins. Rather, the observations support a model, in which axonemal cargoes are stochastically distributed onto the IFT trains.

## Discussion

### ARMC2 is an adapter for IFT of RSs

IFT adapters could be defined as a proteins or protein complexes, which facilitate IFT of a cargo (complex) without being necessary for either IFT itself or the functionality of the cargo once delivered into the cilium. Here, we show that IFT of RSs requires ARMC2 as an adapter. The notion is supported by the following observations: (1) In *pf27* and *armc2* mutants, the presence of RSs was limited to the most proximal region of flagella, (2) IFT of tagged RSs was not observed in *armc2* mutants, and (3) tagged ARMC2 and the essential RS protein RSP3 typically co-migrated on anterograde IFT trains. In comparison to wild-type flagella, the phosphorylation of several spoke proteins is reduced in *pf27/armc2* (*Huang et al., 1981*). While details remain to be explored, it has been proposed that RS phosphorylation occurs near the flagellar tip (*Gupta et al., 2012*). Then, the altered phosphorylation of RS proteins in *pf27/armc2* could result from the RSs failing to reach the flagellar tip because they cannot attach to IFT. We conclude that ARMC2/PF27 ensures efficient delivery of RSs into *Chlamydomonas* flagella and to the flagellar tip by linking RSs to IFT trains.

The transition zone, separating the cell body from the flagellum, functions as a diffusion barrier, which is thought to minimize or prevent the entry of large proteins and complexes into flagella by diffusion (*Garcia-Gonzalo and Reiter, 2012*; *Kee et al., 2012*). This raises the question how residual RSs enter *armc2* flagella. We previously observed that ODAs still accumulated in the proximal region of *ift46* IFT46ΔN flagella, lacking the N-terminal domain of IFT46, which provides the critical docking site for ODA transport via binding of the ODA adapter ODA16 (*Taschner et al., 2017*; *Dai et al., 2018*). Thus, the transition zone of *Chlamydomonas* might allow limited entry of RSs and ODAs in an IFT-independent manner in agreement with studies indicating diffusional entry of large proteins into

primary cilia (*Lin et al., 2013*). Alternatively, ARMC2-independent short-range IFT might shuttle a limited number of RSs through the transition zone into the mutant flagella. However, any mechanism allowing for a slow but steady influx of RSs and ODAs should lead to a significant increase of these complexes in flagella over time, which was not observed. Probably, the transition zone is leakier during flagellar assembly when highly loaded IFT trains enter the flagella or a proper flagellar gate only forms once flagella have reached a certain length, permitting larger complexes to trickle into flagella during early regeneration.

Mutations in mammalian *Armc2* cause sperm malformations resulting in male infertility (*Coutton et al., 2019*). Immunofluorescence analysis of spermatozoa from the affected individuals revealed central pair defects as assessed by the loss of the central pair marker proteins SPAG6 and SPEF2. The spoke head protein RSPH1 was detected in the stunted mutant flagella but the presence of RSs was not further evaluated (*Coutton et al., 2019*). In mammalian cilia, RS defects are often associated with a dislocation or even loss of the central pair apparatus (*Antony et al., 2013*; *Kott et al., 2013*). Thus, a deficiency of the RSs could also lead indirectly to central pair defects. In contrast, *Chlamydomonas pf27* possesses a central pair and transmission electron microscopy and 2D electrophoresis showed that the loss of RSs from most parts of the axoneme is the only apparent defect in this strain (*Huang et al., 1981*; *Alford et al., 2013*). In human patients, *Armc2* defects have been linked only to male infertility, suggesting that Armc2 is expendable for the assembly of motile cilia in the airways and ventricular system (*Coutton et al., 2019*). However, genome-wide interaction studies have linked *Armc2* variants to reduced lung function (*Soler Artigas et al., 2011*; *Pereira et al., 2019*) and Armc2 is highly expressed in ciliated epithelial cells (*Uhlén et al., 2015*). With a length of ~7 μm, airway cilia are comparatively short and it seems possible that RSs enter them over time by diffusion in amounts sufficient to ensure some degree of motility. To conclude, further studies are required to determine if the role of *Chlamydomonas* ARMC2 in RS transport is conserved in other organisms.

## IFT of large axonemal complexes involves adapters

Just like the transport of RSs, IFT of ODAs and IDAs I1/f requires ODA16 and IDA3, respectively, as transport adapters. The flagella of *armc2/pf27*, *ida3*, and *oda16* mutants specifically lack or have greatly reduced amounts of RSs, I1/f dynein, or ODAs, respectively, indicating that these adapter proteins promote transport with single-cargo specificity (*Ahmed and Mitchell, 2005*; *Alford et al., 2013*; *Hunter et al., 2018*). ODA16 also promotes recruitment of ODAs to the flagellar base and ODA transport apparently involves additional factors such as ODA8 and ODA10 (*Dean and Mitchell, 2015*; *Desai et al., 2015*; *Dai et al., 2018*). In conclusion, IFT of three major axonemal substructures, all multiprotein complexes preassembled in the cell body, involves specific adapters suggesting that the transport of other axonemal complexes, such as the other IDAs, may also require adapter proteins.

The identification of ARMC2, the third cargo adapter involved in axonemal assembly, allows us to compare their features: ODA16 and ARMC2 are well conserved in organisms with motile cilia and flagella while IDA3 is not; the latter, however, shows partial similarity to coiled-coil domain-containing protein 24 (*Hunter et al., 2018*). All three proteins were not detected in the original proteomic analysis of fractionated *Chlamydomonas* flagella (Chlamydomonas Ciliary Proteins (chlamyfp.org)) indicating a low abundance in full-length flagella (*Pazour et al., 2005*). Similarly, EST coverage supporting the expression of the genes are very limited or absent (*Albee et al., 2013*). ODA16 is a WD-repeat protein with a small C-terminal intrinsically disordered region, ARMC2/PF27 encompasses armadillo repeats and has an extended N-terminal intrinsically disordered region, and IDA3 possesses several short coiled-coil regions interspersed in a largely intrinsically disordered protein (*Taschner et al., 2017*; *Hunter et al., 2018*). Intrinsically disordered regions often adopt a more defined structure upon binding to their partners (*Dyson and Wright, 2005*). Thus, IFT adapters could fold once they interact with IFT and/or their cargoes. Since the phosphorylated peptides map to the disordered region of ARMC2, phosphorylation in this region could hypothetically regulate ARMC2's folding and molecular interactions (*Johnson and Lewis, 2001*; *Iakoucheva et al., 2004*).

ODA16 interacts with the N-terminal region of IFT46 and this interaction requires both its C-terminal intrinsically disordered region and the WD-repeats whereas the C-terminal region of ODA16 is expandable for ODA binding (*Taschner et al., 2017*). The IFT binding sites of ARMC2/PF27 and IDA3 remain unknown. Potential candidates for IDA3 binding include the IFT-B proteins IFT56 and IFT57 as the flagellar levels of IDAs or IDA subunits are reduced in the corresponding mutants (*Ishikawa*

*et al., 2014*; *Jiang et al., 2017*). Which subunits of the RSs, ODAs, and IDAs interact with the respective adapter and thus mediate transport of the entire complex is currently unknown. In *Drosophila*, RS genes are highly expressed in testis but only the *RSP3/PF14* orthologue *CG32392* is expressed in embryonic chordotonal neurons (*Zur Lage et al., 2019*). Interestingly, the transcript of *CG32668*, the fly orthologue of ARMC2, was abundant in chordotonal neurons but not detected in testis (Andrew Jarman, personal communication, August 2021). The assembly of 9 + 2 sperm flagella by an IFT-independent mechanism likely explains the absence of *Armc2* expression in testis. The expression of both *Armc2* and *CG32392* (*Rsp3*) in chordotonal neurons, which use the IFT pathway to assemble 9 + 0 cilia, could indicate that ARMC2 interacts with RSP3 and that RSP3 is part of these non-motile but mechanosensitive cilia, which possess IDAs and ODAs. Similarly, some RS proteins are also present in certain 9 + 0 motile cilia (*Sedykh et al., 2016*).

## IFT cargo adapters provide an additional level to regulate cargo flux

ARMC2 and IDA3 are enriched in short growing flagella whereas only traces are present in full-length flagella, when cargo transport is contracted. ODA16 is present in full-length flagella (*Ahmed and Mitchell, 2005*) but our analysis showed that is also enriched in growing flagella (*Figure 2—figure supplement 1C*). IFT of ODA16 has not been observed directly but its interaction with IFT46 is well supported by genetic, biochemical, and structural data (*Ahmed et al., 2008*; *Hou and Witman, 2017*; *Taschner et al., 2017*). Thus, it is reasonable to assume that ODA16 moves by IFT and moves more frequently during flagellar regeneration, just as IFT of ODAs is upregulated in growing flagella (*Dai et al., 2018*). ODA16 is present in *oda2* flagella, which lack ODAs (*Ahmed and Mitchell, 2005*; *Hunter et al., 2018*), and IDA3 and ARMC2/PF27 continue to move by IFT in the absence of their respective cargoes (*Hunter et al., 2018*). Further, the transport frequencies of IDA3 and ARMC2/PF27 are still upregulated in short flagella of mutants lacking the respective cargo. In the *armc2 pf14* ARMC2-mS RSP3-NG double-mutant double-rescue strain, numerous ARMC2-mS solo transports were observed, in addition to ARMC2-mS RSP3-NG cotransports. Thus, even when RSs are present, ARMC2 is often loaded onto IFT without its cargo. In a hypothetical model, cells regulate ARMC2-IFT interactions and RSs, when available, will latch on to IFT using the binding sites generated by ARMC2. If correct, the frequency by which these axonemal building blocks are transported into flagella would be controlled to a substantial part by the regulation of IFT-adapter interactions. Binding of the BBSome to IFT trains, for example, is regulated by three small GTPases, IFT22, IFT27, and RabL2 and BBSome-dependent export requires ubiquitination of its GPCR cargoes (*Eguether et al., 2014*; *Desai et al., 2020*; *Shinde et al., 2020*; *Xue et al., 2020*; *Duan et al., 2021*). Thus, both BBSome-IFT and BBSome-cargo interactions appear to be regulated and a similar model could also apply to other adapters including ARMC2. Functional analysis of the phosphorylation sites in the unordered region of ARMC2 could provide insights into the regulation of IFT-ARMC2-RS interactions.

## Unrelated cargoes bind stochastically to IFT trains

Most IFT trains are densely loaded with axonemal cargoes and adapters during early flagellar growth (*Wren et al., 2013*; *Craft et al., 2015*; *Dai et al., 2018*; *Lechtreck et al., 2018*). The transport frequencies decline as flagella approach full length and then, a given cargo is present on only a small subset of the anterograde trains. It is currently unknown whether axonemal cargoes are randomly distributed onto all IFT trains or are preferably transported on a specialized subclass of 'active' or 'open' IFT trains, which are abundant during rapid flagellar growth but scarce during maintenance of full-length flagella. The latter model predicts that unrelated axonemal cargoes will frequently ride on the same IFT trains, which should be particularly apparent in late regenerating flagella when the transport frequencies of axonemal proteins are low. However, the adapters ARMC2 and IDA3 and the ODA subunit IC2 travel apparently independently of each other. If these proteins are representative, cargoes appear to be distributed stochastically onto IFT trains arguing against a subclass of cargo-carrying trains and against a regulation of cargo binding at the level of entire trains. The regulation of IFT-adapter-cargo interaction could instead occur at the level of individual IFT complexes or even independently of IFT trains altogether, for example, by modifications of the adapters and/or cargoes.

Extrapolating our observations on ARMC2, IDA3, and the BBSome, we predict that a random mix of cargoes and adapters decorates the more or less stereotyped IFT backbone of the trains. As cargoes and adapters are substoichiometric to the IFT proteins and specific adapters and cargoes are

present only on a subset of the trains and train units, we predict that each train will possess an individual 'IFT corona', an irregular layer of cargoes and adapters surrounding the repetitive IFT train core; the corona will beprominent during flagellar growth but downsized on trains in full-length flagella (*Figure 6A*).

# Materials and methods

## Key resources table

| Reagent type (species) or resource | Designation | Source or reference | Identifiers | Additional information |
|---|---|---|---|---|
| Genetic reagent (*Chlamydomonas reinhardtii*) | CC-1387, *pf27, mt+* | Chlamydomonas Resource Center | RRID: SCR_014960 | |
| Genetic reagent (*Chlamydomonas reinhardtii*) | CC-613, *pf14, mt-* | Chlamydomonas Resource Center | RRID: SCR_014960 | |
| Genetic reagent (*Chlamydomonas reinhardtii*) | LMJ.RY0402.155726, *armc2 mt-* | Chlamydomonas Resource Center | RRID: SCR_014960 | |
| Genetic reagent (*Chlamydomonas reinhardtii*) | CC-2238 oda16 mt+ | Chlamydomonas Resource Center | RRID: SCR_014960 | |
| Genetic reagent (*Chlamydomonas reinhardtii*) | CC-5412, *ida3:IDA3-NG, mt+* | Chlamydomonas Resource Center | RRID: SCR_014960 | |
| Transfected construct (*Escherichia coli*) | DH10B cells | New England BioLabs | – | Competent cells |
| Antibody | Rabbit anti-Sheep IgG (H + L) Secondary Antibody, HRP | Thermo Fisher | Catalog #: 31480. http://antibodyregistry.org/AB_228457 | WB 1:2000–5000 |
| Antibody | **Mouse IgG (H + L) Cross-Adsorbed Secondary Antibody** | Thermo Fisher | Catalog #: 31432. http://antibodyregistry.org/AB_228302 | WB 1:2000–5000 |
| Antibody | IgG (H + L) Goat anti-Rat, HRP, Invitrogen | Thermo Fisher | Catalog #: 31470. http://antibodyregistry.org/AB_228356 | WB 1:2000–5000 |
| Antibody | Goat anti-mouse IgG (H + L) Alexa Fluor 488 (mouse polyclonal) | Invitrogen | Catalog #: A-11029. RRID: AB_2534088 | IF 1:800 |
| Antibody | Goat anti-rabbit IgG (H + L) Alexa Fluor 568 (rabbit polyclonal) | Invitrogen | Catalog #: 11,036. RRID: AB_10563566 | IF 1:800 |
| Antibody | Anti-HA, High Affinity (rat monoclonal, clone 3F10) | Roche/Sigma | Catalog #: 11867423001 | WB 1:800 |
| Recombinant DNA reagent | pGEMT-ARMC2(–3xTAG/mS) plus Hyg | This paper | | Expression vector encompassing the up- and downstream flanking regions of the ARMC2 gene, optional mS or 3xTAG (NG-3xHA-6xHis) epitope tags and the aph7" selectable marker gene. Available from the corresponding authors. |
| Sequence-based reagent | S1** | This paper | PCR Primer | CCGCCTGCACCCTTATC GCTGCCTCTGTCCCTCTTCC |
| Sequence-based reagent | AS2 | This paper | PCR Primer | CCTGTTCCGCACGCTG GTCTACCGTCTACC |
| Sequence-based reagent | S3* | This paper | PCR Primer | CGAGGCGGTGAGCGAGCA CGTGTTCCGACTCATG |
| Sequence-based reagent | AS3* | This paper | PCR Primer | GCCTCACGGTACCGTGAGC ACATGCATGGGTTTGC |
| Sequence-based reagent | S4 | This paper | PCR Primer | CGCAACCCCCGCTAC TCTAACCTCGAGG |

*Continued on next page*

*Continued*

| Reagent type (species) or resource | Designation | Source or reference | Identifiers | Additional information |
|---|---|---|---|---|
| Sequence-based reagent | AS4Hind | This paper | PCR Primer | CAGAAGCTTGAAGCCCG AAAGCTGACGAAGTGGG |
| Sequence-based reagent | HindS6.1 | This paper | PCR Primer | GAGAAGCTTACCTACCTGG GTCTTGACATGCCCTGTCC |
| Sequence-based reagent | AS5Xho | This paper | PCR Primer | CCTCGAGCTCCGGCAA CGCCTCCAGCTCC |
| Sequence-based reagent | XhoS7 | This paper | PCR Primer | CCTCGAGTAGGGGCCTT GCTTAGGGAATTCAGGG |
| Sequence-based reagent | AS6 | This paper | PCR Primer | CTCGCTTTCACAACTCC AGGGTGCCCATGC |
| Sequence-based reagent | ida3f | This paper | PCR Primer | ATTTGGACGGA GCCTTGAC |
| Sequence-based reagent | ida3r | This paper | PCR Primer | TGTTTCGCACG CCTTCA |
| Chemical compound, drug | ProLong Gold Antifade Mountant | Thermo Fisher | Catalog #: P36934. RRID:SCR_015961 | Catalog number #: P36930 |

## Strains and culture conditions

The previously described *pf27* (CC-1387) and *pf14* (CC-613) mutant strains and the CliP strains LMJ.RY0402.083979 and LMJ.RY0402.155726 (i.e., *armc2*) are available from the *Chlamydomonas* Resource Center (RRID:SCR_014960). The *pf14* RSP3-NG, IDA3-NG, the ida3 IDA3-NG (CC-5412), and the *oda3 oda6* IC2-NG strains were described previously (*Zhu et al., 2017*; *Dai et al., 2018*; *Hunter et al., 2018*). The *armc2 pf14* ARMC2-mS RSP3-NG, *armc2 oda3 oda6* ARMC2-mS IC2-NG, and *ida3 armc2* IDA3-NG ARMC2-mS strains were generated by mating and the *armc2 pf14* ARMC2-3xTAG strain by transforming the ARMC2-3xTAG plasmid into a *pf14 armc2* double mutant obtained by mating. Progeny with the desired combination of alleles were identified using a combination of selection (CliP mutants are resistant to paromomycin), PCR, fluorescent imaging, and Western blotting. The *ida3* mutation generates an SfcI site and we used the primers ida3f (ATTTGGACGGAGCCTTGAC) and ida3r (TGTTTCGCACGCCTTCA) to amplify the genomic region flanking this site followed by restriction digest with Sfc1 to track the *ida3* mutant allele (*Hunter et al., 2018*).

Cells were maintained in M medium (https://www.chlamycollection.org/methods/media-recipes/minimal-or-m-medium-and-derivatives-sager-granick/) at a 14:10 hr light:dark cycle at 24°C. Cells maintained in unaerated flask for imaging, transformation, and phenotypical analysis; for flagellar isolation, cultures were aerated with air supplemented with 0.5% $CO_2$.

## ARMC2 cloning

The cloning scheme for the 13.6 kB ARMC2 genomic DNA is depicted in the supplemental data (*Figure 1—figure supplement 1C*). Briefly, using PCR and purified genomic DNA as a template, five genomic DNA fragments of the ARMC2 gene were amplified. Fragment 1 of 4.6 kB, including 1 kB 5' flanking sequence, was amplified with primer S1** and AS2 and cloned into the pGEM-T vector by AT cloning. Fragment 2 of 3.4 kB and fragment 3 of 3.1 kB were amplified using primer pairs S3*/AS3* and S4/AS4 HindIII, respectively, AT cloned and then combined into one plasmid by releasing fragment 3 with a XhoI/SacI digest and ligating it into the pGEM-T-fragment 2 plasmid digested with the same enzymes. This resulted in a plasmid with the 6.5 kB fragment 2' segment. The HindIII restriction sequence introduced into an intron by the primer was used for subsequent ligation with the downstream fragment. Similarly, fragments 4 and 5 were amplified using primer pairs HindS6.1/AS5Xho and XhoS7/AS6 and AT cloned into pGEM-T. An XhoI restriction sequence was added into the primers to ligate fragment 5, released by an XhoI and SacII digest, with fragment 4 in the pGEM-T plasmid digested with the same enzymes. This resulted in the plasmid containing the 2.5 kB fragment 3'. The tag (3xTAG or mS) was inserted into the Xho1 site of fragment 3' and the correct orientation was verified by restriction digest. Then, three fragments, including fragments 2' and 3', were released by Spe1 and HindIII and HindIII and NdeI digests, respectively, and together with a 1.7 kB fragment

conferring the hygromycin (Hyg) resistance (*Zhu et al., 2017*), inserted into the pGEMT-fragment 1 plasmid digested with SpeI and NdeI. The ligation mixture was transformed into DH10B competent *Escherichia coli* cells (New England Biolab, MA). The final pGEM-T-ARMC2 construct was confirmed by restriction digest. Purified plasmids of the ARMC2 genomic construct were transformed into *pf27* and *armc2* cells using the glass beads method (*Kindle, 1990*; *Zhu et al., 2017*). Transformants were selected on plates containing 10 µg/ml of hygromycin and the resistant clones were suspended in 10 mM HEPES and further screened for motility and fluorescence.

## Flagellar isolation and Western blotting

For Western blot analyses of flagella, cells were washed in 10 mM HEPES, resuspended in 10 mM HEPES, 5 mM MgSO$_4$, 4% sucrose (w/v), and deflagellated by the addition of dibucaine. After removing the cell bodies by two differential centrifugations, flagella were sedimented at 40,000× *g*, 20 min, 4°C as previously described (*Witman, 1986*). To obtain short regenerating flagella, cells were deflagellated by a pH shock, transferred to fresh M medium, stored on ice for ~15 min, diluted with M medium, and allowed to regenerated flagella for ~18 min in bright light with agitation. Then, cells were sedimented, washed once in 10 mM HEPES and deflagellated as described above. Flagella were dissolved in Laemmli SDS sample buffer, separated on Mini-Protean TGX gradient gels (BioRad), and transferred electrophoretically to PVDF membrane. After blocking, the membranes were incubated overnight in the primary antibodies; secondary antibodies were applied for 90–120 min at room temperature. After addition of the substrate (Femtoglow by Michigan Diagnostics or ECL Prime Western Blotting Detection Reagent by GE Healthcare), chemiluminescent signals were documented using a BioRad Chemi Doc imaging system. The following primary antibodies were used in this study: rabbit anti-RSP3 (1:800) and, for immunofluorescence, affinity-purified anti-RSP3 (1:100; *Williams et al., 1986*), rabbit anti-NDK5 (1:1000; *Chung et al., 2017*), rabbit anti-IFT54 (1:800; *Wingfield et al., 2017*), rabbit anti-NAB1 (1: 5000; Agrisera), rabbit anti-GFP (A-11122; Thermo Fisher), mouse monoclonals anti-IFT139 (1:400; *Cole et al., 1998*), anti-IC1 (1:5000; *King and Witman, 1990*), and anti-IC2 (1:2000 for Western blotting and 1:500 for immunofluorescence; *King and Witman, 1990*), and rat monoclonal anti-HA (1:800, clone 3F10 Roche/Sigma).

## Indirect immunofluorescence

For indirect immunofluorescence, cells were sedimented, resuspended in HMEK, allow to settle onto polyethylenimine (0.2%) coated multiwell slides for 1–2 min and submerged into –20°C methanol for 8 min. The slides were air-dried, blocked (1% BSA in PBS-T), washed with PBS, incubated with the primary antibodies in blocking buffer overnight, washed, stained with secondary antibodies (1:800 Alexa Fluor anti-rb-565 and anti-mo-488; Invitrogen), washed in PBS-T, submerged briefly in 80% ethanol, air-dried, and mounted in ProlongGold (Invitrogen).

For wide-field epifluorescence microscopy, images were taken using a 60 × 1.49 objective, an Eclipse Ti-U microscope (Nikon) equipped with a Lumen200 light source (PRIOR) and filters for FITC and TexasRed. Alternatively, images were taken using a 40× Plan Fluor lens on an Eclipse E600 microscope (Nikon) equipped with a DFC9000 GT sCMOS camera and the accompanied imaging software (Leica, Wetzlar, Germany). The bandpass of filters was Ex: 460–500 nm, Dm: 505 nm, Em: 510–560 nm for NeonGreen (NG) and Ex: 543–568 nm, Dm: 570 nm, Em: 579–612 nm for mSc. The illuminator was a four-channel SOLA SM 365 LED light engine with excitation peaks at 365, 470, 530, 590 nm (Lumencor, Beaverton, OR).

## Live cell microscopy

For TIRF imaging, we used an Eclipse Ti-U microscope (Nikon) equipped with 60× NA1.49 TIRF objective and through-the-objective TIRF illumination provided by a 40 mW 488 nm and a 75 mW 561 nm diode laser (Spectraphysics) as previously described (*Lechtreck, 2013*; *Lechtreck, 2016*). The excitation lasers were cleaned up with a Nikon GFP/mCherry TIRF filter cube and the emission was separated using an Image Splitting Device (Photometrics DualView2 with filter cube 11-EM) supplemented with an et595/33m filter to reduced chlorophyll autofluorescence and a meniscus lens to adjust for focus. Images were recorded at 10 fps using an iXON3 (Andor) and the NIS-Elements Advanced Research software (Nikon). The optical set-up for the focused laser beam used for FRAP analysis was previously described by *Wingfield et al., 2017*. FIJI (National Institutes of Health) was used to

generate kymograms using either the build-in Multi Kymogram tool or the KymoResliceWide plugin (https://imagej.net/KymoResliceWide). The Plot Profile tool was used to analyze signal intensity and Microsoft Excel was used for statistical analysis. Adobe Photoshop was used to adjust image contrast and brightness, and figures were prepared in Adobe Illustrator.

Observation chambers for *Chlamydomonas reinhardtii* were constructed by applying a ring of vacuum grease or petroleum jelly to a $24 \times 60$ mm$^2$ No. 1.5 coverslip; 10 µl of cell suspension were applied and allowed to settle for ~1 min. Then, the chamber was closed by inverting a $22 \times 22$ mm$^2$ No. 1.5 cover glass with ~5–10 µl of 5 mM HEPES, pH 7.3 supplemented with 3–5 mM EGTA onto the larger cover glass. Cells were imaged through the large cover glass at room temperature. Regenerating cells were obtained as follows: Cells in fresh M medium were deflagellated by a pH shock, sedimented, resuspended in a small volume of M medium, and stored on ice for 15 min or until needed. Then, cells were diluted with M medium and agitated in bright light at room temperature. Aliquots were analyzed by TIRF microscopy at various time points. To generate long-short cells, cells in M medium were chilled on ice for ~3 min and then passed repeatedly (5–8×) through a 26 G½ needle using a 1 ml syringe (*Ludington et al., 2012*). The presence of spinning cells, indicative or the loss of just one flagellum, was verified by microscopy and the cells we allowed to regenerate for 5–10 min prior to mounting for TIRF microscopy.

To determine the swimming velocity, cells in fresh M medium were placed in a chambered plastic slide (Fisherbrand, 14-377-259) and observed using an inverted cell culture microscope (Nikon,TMS). Using a constant exposure time of 1 s, images were taken using a MU500 camera (Amscope) and Topview software. The length of the trajectories resulting from the cells' movements was analyzed using ImageJ and converted into µm/s.

## Probability of cotransport calculation

For our calculations, we assumed an IFT train frequency of 60/min, which is close to that reported in several studies of IFT in *Chlamydomonas* (*Kozminski et al., 1993*; *Dentler, 2005*; *Wingfield et al., 2017*). The probability of ARMC2-mS transport ($P_{(ARMC2-mS)}$) in our data set is 0.268 (=243 of 905 trains carried ARMC2-mS) and that for IDA3-NG transport ($P_{(IC2-NG)}$) is 0.117. Multiplication of these two probabilities results in 0.03, as the expected probability ($P_{cotransport-calculated}$) of ARMC2-mS and IDA3-NG being cotransported if they bind independently of each other to IFT trains. This value is close to 0.022, the observed probability of cotransports ($P_{cotransport-observed}$; see *Table 1*). The probability of ARMC2-mS/RSP3-NG and ARMC2-mS/IC2-NG cotransports was calculated similarly.

## Acknowledgements

We thank Yuqinq Hou and George Witman (UMASSMED) for sharing their proteomic data on ARMC2, Gui Zhang and Stephanie Chen for technical support, Win Sale (Emory University) for critical reading of the manuscript, and Paul Lefebvre and Matt Laudon (*Chlamydomonas* Resource Center) for the advice using on the CLiP library. This study was supported by grants by the National Institutes of Health (R01GM110413 to KL and R015GM128130 to PY and LMA). The content is solely the responsibility of the authors and does not necessarily represent the official views of the National Institutes of Health.

## Additional information

### Funding

| Funder | Grant reference number | Author |
| --- | --- | --- |
| National Institute of Health | R01GM110413 | Karl F Lechtreck |
| National Institute of Health | R015GM12813 | Lea Alford<br>Pinfen Yang |

The funders had no role in study design, data collection and interpretation, or the decision to submit the work for publication.

## Author contributions
Karl F Lechtreck, Conceptualization, Formal analysis, Funding acquisition, Investigation, Supervision, Visualization, Writing - original draft, Writing - review and editing; Yi Liu, Investigation, Methodology; Jin Dai, Rama A Alkhofash, Jack Butler, Investigation; Lea Alford, Funding acquisition, Project administration, Writing - review and editing; Pinfen Yang, Conceptualization, Funding acquisition, Investigation, Supervision, Writing - review and editing

## Author ORCIDs
Karl F Lechtreck http://orcid.org/0000-0002-6219-6470
Pinfen Yang http://orcid.org/0000-0002-3773-0053

## Decision letter and Author response
Decision letter https://doi.org/10.7554/eLife.74993.sa1
Author response https://doi.org/10.7554/eLife.74993.sa2

## Additional files

### Supplementary files
• Transparent reporting form

### Data availability
All data generated or analysed during this study are included in the manuscript and supporting file; Source Data files have been provided for the western blots in Figures 1, 2, Figure 1 -Supplement 1 and Figure 2 - Supplement 1.

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
