## [Editor Report]

This paper is of broad interest to readers interested in motile cilia and cargo transport mediated by intraflagellar transport (IFT). It examines how radial spokes are trafficked into cilia by IFT, which represents a key process in the assembly of motile cilia. The authors demonstrate that an adaptor protein (ARMC2) is needed for association of radial spokes with the IFT machinery. They also find that three distinct axonemal proteins and adapters interact in a stochastic manner with individual IFT trains (or particles) rather than being transported by a specialized subset of trains specifically designated for axonemal proteins. The results support the key claims in the paper.

---

## [Decision Letter]

**Decision letter after peer review:**

Thank you for submitting your article "*Chlamydomonas* ARMC2/PF27 is an obligate cargo adapter for IFT of radial spokes" for consideration by *eLife*. Your article has been reviewed by 2 peer reviewers, and the evaluation has been overseen by a Reviewing Editor and Anna Akhmanova as the Senior Editor. The reviewers have opted to remain anonymous.

Essential revisions:

1) Page 12 bottom paragraph: the authors note that in cultures of armc2 and pf27, they occasionally find motile cells that over the course of a few days would become fully motile with incorporated radial spokes. Although further experimental evidence on this is not required, it might help the reader if they could indicate how frequently this phenomenon was observed and at least offer thoughts on possible ways in which this might arise. If the observations are very infrequent and the authors are unable to explain them, they might consider removing this paragraph from the text.

2) The authors propose that RS localization in the proximal region of pf27 flagella occurs by diffusion of RS from the cell body. This would seem to imply a certain leakiness to the ciliary gate even for large mega-dalton sized complexes. The authors need to discuss this result in the context of studies done in e.g. mammalian cells, where diffusion from the cytosol of soluble proteins larger than 50 kDa was found (in some studies) to be inhibited by the ciliary gate/transition zone (PMID 26472341). Further, is it possible that there is an additional (unknown) IFT cargo adapter for RS that promotes RS transport across the ciliary gate? Finally, the authors suggest that IFT-adaptor interactions are critical for control of cargo entry into cilia. While that certainly seems quite reasonable, control of the adaptor-cargo association is likely equally important for productive transport, off-loading and axonemal incorporation.

---

## [Author Response]

Essential revisions:1) Page 12 bottom paragraph: the authors note that in cultures of armc2 and pf27, they occasionally find motile cells that over the course of a few days would become fully motile with incorporated radial spokes. Although further experimental evidence on this is not required, it might help the reader if they could indicate how frequently this phenomenon was observed and at least offer thoughts on possible ways in which this might arise. If the observations are very infrequent and the authors are unable to explain them, they might consider removing this paragraph from the text.

Currently, we cannot explain how *armc2* and *pf27* cells regain motility. The frequency by which this “suppression/reversion (?)” occurred varied depending on the genetic background (observed more often in some *armc2* an *pf27* progeny than others) and apparently, the nutritional state of the cells. As the phenomenon is rare, statistical data on this complex phenomenon are not yet available. Also, a key tool, an antibody directed against ARMC2, is not yet available, preventing us from analyzing whether such motile *armc2/pf27* cells have regained the ability to express ARMC2, e.g., by gaining the ability to remove the large intron containing the insertion cassette from the *armc2* mRNA by splicing.

We feel that it is important to inform the reader of this phenomenon, as we plan to explore this further in future. We moved a shorter version of this observation to the first paragraph of the results.

“Rarely, motile cells emerged in armc2 and pf27 cultures and, over a few days, the number of such cells increased, motility improved and the amount of RSP3 in flagella increased (not shown). The phenomenon was not further explored in this study.”

2) The authors propose that RS localization in the proximal region of pf27 flagella occurs by diffusion of RS from the cell body. This would seem to imply a certain leakiness to the ciliary gate even for large mega-dalton sized complexes. The authors need to discuss this result in the context of studies done in e.g. mammalian cells, where diffusion from the cytosol of soluble proteins larger than 50 kDa was found (in some studies) to be inhibited by the ciliary gate/transition zone (PMID 26472341). Further, is it possible that there is an additional (unknown) IFT cargo adapter for RS that promotes RS transport across the ciliary gate?

The mechanism by which the residual RSs enter *armc2* flagella is currently unknown. Previously, we observed that outer dynein arms (as visualized by mNeonGreen-tagging of the essential IC2 subunit) similarly accumulate in the very proximal region of flagella of *ift46* IFT46ΔN cells, which lack the N-terminal domain of IFT46 critical for ODA transport. This suggests that residual amounts of these large complexes can pass through the ciliary gate without using the docking sites established for long-distance IFT to the flagellar tip, if not by an IFT-independent manner altogether. However, if the transition zone, which typically is a barrier for complexes above a ~50 kD threshold, is leaky for RSs and ODAs, one would expect their levels to progressively increase with time and after several hours, approach levels seen in control, which was not observed. Probably, entry of such large complexes into the flagellum is facilitated during flagellar growth, when IFT trains highly loaded with structural proteins move through the transition zone. It is also possible that a mature flagellar gate is only formed after the assembly of flagellar stubs allowing flagellar entry of large complexes during the early stage of flagellar generation,

We added the following to the Discussion section of the manuscript:

“The transition zone, separating the cell body from the flagellum, functions as a diffusion barrier, which is thought to minimize or prevent the entry of large proteins and complexes into flagella by diffusion (Garcia-Gonzalo and Reiter 2012; Kee et al. 2012). […] Probably, the transition zone is leakier during flagellar assembly when highly loaded IFT trains enter the flagella or a proper flagellar gate only forms once flagella have reached a certain length, permitting larger complexes to trickle into flagella during early regeneration.”

Finally, the authors suggest that IFT-adaptor interactions are critical for control of cargo entry into cilia. While that certainly seems quite reasonable, control of the adaptor-cargo association is likely equally important for productive transport, off-loading and axonemal incorporation.

We agree. In the original manuscript (p23) we stated: “The model does not exclude that adapter-cargo interactions are regulated as well.” However, this statement might get lost in our discussion of IFT-adapter interaction. Therefore, we reworded the paragraph as follows:

“If correct, the frequency by which these axonemal building blocks are transported into flagella would be controlled to a substantial part by the regulation of IFT-adapter interactions. […] Functional analysis of the phosphorylation sites in the unordered region of ARMC2 could provide insights into the regulation of IFT-ARMC2-RS interactions.”